# Improvements in Wear and Corrosion Resistance of Ti-W-Alloyed Gray Cast Iron by Tailoring Its Microstructural Properties

**DOI:** 10.3390/ma17102468

**Published:** 2024-05-20

**Authors:** Abdul Razaq, Peng Yu, Adnan Raza Khan, Xiao-Yuan Ji, Ya-Jun Yin, Jian-Xin Zhou, Taher A. Shehabeldeen

**Affiliations:** 1State Key Laboratory of Materials Processing and Die & Mould Technology, Huazhong University of Science and Technology, Wuhan 430074, China; abdulrazaqumt@yahoo.com (A.R.); d202280419@hust.edu.cn (P.Y.); jixiaoyuan@hust.edu.cn (X.-Y.J.); 2Department of Engineering, University of Technology and Applied Sciences, Muscat 133, Oman; adnan.khan@utas.edu.om; 3Mechanical Engineering Department, Faculty of Engineering, Kafrelsheikh University, Kafrelsheikh 33516, Egypt; taher_atia@eng.kfs.edu.eg

**Keywords:** gray cast iron, tungsten and titanium additives, microstructural properties, mechanical properties, wear and corrosion resistance

## Abstract

The improved wear and corrosion resistance of gray cast iron (GCI) with enhanced mechanical properties is a proven stepping stone towards the longevity of its versatile industrial applications. In this article, we have tailored the microstructural properties of GCI by alloying it with titanium (Ti) and tungsten (W) additives, which resulted in improved mechanical, wear, and corrosion resistance. The results also show the nucleation of the B-, D-, and E-type graphite flakes with the A-type graphite flake in the alloyed GCI microstructure. Additionally, the alloyed microstructure demonstrated that the ratio of the pearlite volume percentage to the ferrite volume percentage was improved from 67/33 to 87/13, whereas a reduction in the maximum graphite length and average grain size from 356 ± 31 µm to 297 ± 16 µm and 378 ± 18 µm to 349 ± 19 µm was detected. Consequently, it improved the mechanical properties and wear and corrosion resistance of alloyed GCI. A significant improvement in Brinell hardness, yield strength, and tensile strength of the modified microstructure from 213 ± 7 BHN to 272 ± 8 BHN, 260 ± 3 MPa to 310 ± 2 MPa, and 346 ± 12 MPa to 375 ± 7 MPa was achieved, respectively. The substantial reduction in the wear rate of alloyed GCI from 8.49 × 10^−3^ mm^3^/N.m to 1.59 × 10^−3^ mm^3^/N.m resulted in the upgradation of the surface roughness quality from 297.625 nm to 192.553 nm. Due to the increase in the corrosion potential from −0.5832 V to −0.4813 V, the impedance of the alloyed GCI was increased from 1545 Ohm·cm^2^ to 2290 Ohm·cm^2^. On the basis of the achieved experimental results, it is suggested that the reliability of alloyed GCI based on experimentally validated microstructural compositions can be ensured during the operation of plants and components in a severe wear and corrosive environment. It can be predicted that the proposed alloyed GCI components are capable of preventing the premature failure of high-tech components susceptible to a wear and corrosion environment.

## 1. Introduction

The surface deformation of GCI by wear and corrosion are the key factors which inhibit its required service life. To combat this problem, tailoring the microstructural properties of mass-produced GCI with alloying elements to control the desired properties, such as wear and corrosion resistance, has been considered as an effective approach to ensure the smooth functionality of systems. It has been observed that the addition of alloying elements increases the pearlite-to-ferrite ratio and refines the grain size. All such factors contribute to improving the mechanical properties, which results in the enhancement of the corrosion and wear resistance of GCI [1,2,3].

GCI is mainly composed of A-type graphite flakes along with a variable volume fraction of pearlite and ferrite, which mostly controls its wear and corrosion properties [4]. The small size of graphite flakes and higher pearlite-to-ferrite ratio contribute not only to enhancing wear resistance but also results in higher mechanical strength and hardness [5]. The improvement in wear resistance can be attributed to the morphology of graphite flakes and the formation of a graphite film due to abrasion on the wear surface [6]. The higher mechanical hardness is based on the higher volume percentage of pearlite, which can resist surface deformation during the wear phenomena [7,8]. Therefore, it is established that the modified microstructure of GCI with a fine grain size, small amount of graphite, and higher pearlite-to-ferrite ratio is the main requirement for achieving better wear resistance [9,10,11].

As it is described, the microstructure of GCI is a heterogeneous combination of graphite flakes, pearlites, and ferrites. The heterogeneous microstructure generates cathodic and anodic sites, which can cause electromotive force with a certain potential difference and promote graphitic corrosion [12]. Due to the penetration of electrolyte within a metallic structure through graphite flakes, corrosion is formed, which results in the splitting of iron, leaving behind a porous mass of graphite. However, metallic microstructures with a fine grain size and higher pearlite-to-ferrite volume fraction ratio exhibit good corrosion resistance [13]. Microstructure morphology contains graphite–ferrite couples adjacent to each other. The main cause of corrosion initiation is the electromotive force build up by graphite–ferrite couples, which facilitates electrolyte penetration into the metal matrix along with graphite flakes [13]. Cementite is a constituent of pearlite, which has a higher electrode potential compared to ferrite. Therefore, its small grain size and uniform distribution with the higher volume fraction of pearlite support the achievement of higher values of cementite and increases in mechanical hardness [14]. Therefore, the scarcity of ferrite can help to inhibit corrosion attack.

Ti and W, well-known hard carbide-forming alloying elements of GCI, change the microstructure of GCI and improve its mechanical and other functional properties. Ti is very reactive to form TiO_2_ layer, which can passivate the surface of GCI and provide protection against a corrosion attack, especially in seawater applications [15]. The addition of the alloying element Ti tailored the microstructure of GCI and resulted in the promotion of the pearlite-to-ferrite ratio, the significant refinement of the flakes with the nucleation of D-type graphite flakes, and improved the wear resistance [16]. It has also been observed that alloying with Al, Ti, and Zr modifies the microstructure of GCI with the nucleation of D + E-type graphite flakes [17]. A comprehensive literature review shows that Ti alloy in the GCI metal matrix has improved its wear and corrosion resistance, but microstructure and mechanical variations were not reported. Tungsten (W) is also a potential alloying element for pearlite promotion [18] and mechanical hardness improvements in cast iron, resulting in improvements in the wear and corrosion resistance [19]. Moreover, it was found in the literature that fine grains of WC (tungsten carbide) contribute to the formation of hard carbides, the enhancement of mechanical properties, and wear and corrosion resistance [20]. Keeping in view the literature survey, it was found that Ti-W alloy plays a vital role in enhancing the mechanical properties of GCI, which also improves wear and corrosion resistance.

The combination of Ti-W alloying elements has been used in Fe-based composite prepared by a ball milling process for the modification of the microstructure, and an improvement in the wear resistance is observed when using a sintering temperature technique. Nevertheless, the underlying methodology has a limitation in terms of being exercised on an industrial scale due to the involvement of a complex synthesis process. Another approach to improve the wear and corrosion resistance of GCI is the coating of TiC and WC at the surface of GCI [21,22,23]. The surface coating technique has a limitation in terms of being applied in complex geometrical structures, and a long amount of time is required in the process of its mass production [1]. Laser treatment for the surface hardening of GCI has been demonstrated for the improvement in its mechanical properties and wear and corrosion resistance [24,25,26]. But the laser treatment process has the adverse effects of generating a non-homogenous phase transformation in a metal matrix, and this technique can only be applied on a small scale instead of in mass production [27]. The evolution of the GCI microstructure is observed by a heat treatment process, which improved its mechanical properties and wear resistance [28,29,30]. The major disadvantages of this process were found to be that the heat treatment cycle was relatively lengthy and it required equipment maintenance and high operational costs [31]. The improvement in the wear resistance of the austenitic steel metal matrix composite is observed by the reinforcement of (Ti, W)C [32], whereas the microstructural modifications in the steel metal matrix by (T, W)C reinforcement and its correlation with both wear and corrosion resistance still need to be studied.

From our literature survey, we could not find a comprehensive research study simultaneously related to the wear and corrosion response by tailoring the microstructure of Ti-W-alloyed GCI by using a conventional casting production method on an industrial scale. With reference to the above limitations, the proposed method is based on charge melting by using a conventional induction furnace, and a casting process was found to be the most robust technological solution for the industrial-scale production of microstructurally modified GCI. Therefore, the aim of this experimental work was to modify the microstructure of GCI by alloying Ti-W in order to obtain the required mechanical properties, which caused improvements in the wear and corrosion resistance. This work may enable design engineers to select the appropriate compositions of materials to achieve the required performance of GCI working in a wear and corrosion environment during the specific service life span.

GCI has many industrial applications that initially focused on the manufacturing of heavy vehicle parts such as tractors, trucks, and cylinder heads for diesel engines [33]. Recently, there has been an increasing trend, especially in the automotive sector, towards the production of cylindrical blocks to achieve the high performance of diesel engines by improving their wear and corrosion properties [34]. GCI is considered suitable to bear small mechanical loads, owing to its lower tensile strength. Moreover, GCI is considered to survive in medium wear and corrosion environmental conditions. To overcome these shortcomings, the alloyed GCI can fulfil user requirements cost-effectively. The continuous use of GCI in wear and corrosion environments requires its regular maintenance and service; consequently, the span of its required service life is shortened. The sustainability of the alloyed GCI to withstand high mechanical loads can be enhanced. Therefore, the alloyed GCI with improved wear and corrosion resistance can be used for long durations in severe environments [35]. Moreover, the service life of the components of alloyed GCI, such as a brake disk used in a wear-and-tear environment, can be enhanced. Similarly, the service life of alloyed GCI parts such as water pipes and valves can also be improved [36].

This work investigates the correlation between the microstructural modifications of Ti-W-alloyed GCI and variations in the mechanical properties and wear and corrosion resistance by using various characterization techniques. We have evaluated the macroscopic structure of GCI with and without Ti-W alloying elements through an optical microscope to identify the formation of graphite flakes and improvements in its mechanical properties, such as a tensile test, yield test, and macro Vickers hardness test. The microscopic and chemical composition analysis is performed by using scanning electron microscopy and electron backscatter diffraction techniques. The wear resistance of the alloyed and pristine GCI is characterized by pin-on-disc tribo-tests, and the surface roughness of the wear track is measured by the interferometric technique, respectively. The corrosion resistance of both the alloyed and pristine GCI is assessed under the seawater condition by using potentiodynamic polarization and electrochemical impedance spectroscopy techniques. The implemented experimental approaches helped us to understand the correlation between microstructural modifications of GCI by alloying Ti-W elements, as well as to comprehend the wear and corrosion resistance properties easily.

## 2. Experimental Procedure

In order to cast the tensile test samples, the green sand molds’ composition consists of 63.5% green sand, 20% bentonite, 12% coal dust, 1.5% yellow dextrin, and 3% water. The green sand mold composition was mixed for 4 min in a sand muller. The molds for the GCI specimens were prepared using a mild steel molding box of the desired dimensions as per the ASTM standard E8/E8M [37]. The green sand molds were then coated with an ethanol-based (isopropyl alcohol) coating [38] and baked at 200 °C for 8 min under the CO_2_ gas flow. After baking, the molds were cooled to room temperature and cleaned with high-pressure air [39].

Gray cast iron was produced in the high-frequency induction furnace with a 10 Kg capacity and an acidic refractory acid lining. The base metal of GCI was reinforced with Ti (purity: 99.9%, particle size: 22.4 μm) and W (purity: 99.9%, particle size:14.5 μm) in powder form as an alloying element. Ferrosilicon contains 60% silicon content and has a particle size less than 3 mm, and it was used as an inoculant in a quantity equal to 0.5% of the total charge.

The gray cast iron, labeled as Sample B-1, was first produced in an induction furnace and held inside the furnace at 1450 °C for 15 min. The molten metal was then tapped into a pre-heated ladle (i.e., reddish in appearance) and stirred continuously at 1400 °C to ensure the homogeneity of the inoculant. Then, it was vertically poured into a mold at 1350 °C. Sample B-1 served as the base matrix material for the preparation of Samples B-2, B-3, and B-4. The gradual increase in the proportion of Ti and W was the main criteria involved in Samples B-2, C-3, and D-4, where Sample B-1 contains a lower amount of Ti and W, Sample B-2 contains an intermediate amount of Ti and W, while Sample B-3 contains a higher amount of Ti and W, as shown in Table 1. Sample B-2 of gray cast iron was heated to 1550 °C and the required amount of Ti-W powder was added and held inside the furnace for 15 min. After that, it was tapped into a pouring ladle at 1520 °C and gently stirred to ensure the homogeneity of the inoculant. Finally, the molten gray cast iron from the ladle was vertically poured into a mold, similar to that of Sample B-1. The total time from tapping to pouring was about 10 min for all the samples. The same procedure was repeated for Sample B-3 and Sample B-4 as well.

Tensile testing was performed with the help of a computer-controlled universal testing machine (servo-hydraulic MTS 810, Patras, Greece) at a loading rate of 10 mm·min^−1^. The ultimate tensile strength and yield strength were calculated by taking the average of the three measurements. The dimensions of the sample for tensile testing are shown in Figure 1. The scanning electron microscope (SEM) characterization technique was implemented to investigate the surface morphologies of all the fractured tensile test samples at 25 °C.

For the microstructure analysis, round disc-shaped samples of thicknesses of 20 mm and diameters of 30 mm were manually ground with SiC papers of up to grade 3000 and polished with diamond paste of size 2.5 µm and 0.5 µm using a velvet cloth according to the ASTM E003-11 standard [40]. Optical microscope model# MR-6000 (OM, DSX-HRSU, Olympus, Tokyo, Japan), SEM model#JSM-6510 (Hitachi, Tokyo, Japan) and X-ray diffractometer model# Shimadzu-7000 (Columbia, MD, USA) were used for the microstructural analysis of the GCI samples. Electron backscattered diffraction (EBSD) is utilized to evaluate the distribution of Ti and W alloying elements as well as to check the grain size distribution and orientation. Image-J software (Version 1.54i 03) was used to determine the grain size and the volume fraction of ferrite and pearlite. The elemental compositions of the microstructure of gray cast iron in the selected areas were measured by a spark emission spectrometer equipped with SEM. For Brinell hardness testing, disc-shaped samples were prepared with a thickness of 10 mm and a diameter of 15 mm in compliance with the TS EN 6506-4 201 standard [41]. The surfaces of all the samples were ground on SiC papers of up to grade 2000 to eliminate any lingering debris. The Brinell hardness test was performed with a steel ball of 10 mm diameter under a load of 3000 kg.

The electrochemical properties of GCI were investigated in 3.5 wt.% NaCl solution. Samples were cold-mounted using epoxy resin; one side of exposed sample with a dimension of 1 cm^2^ was immersed into the electrolyte. The corrosion kinetics, mechanism, and polarization behavior of GCI were analyzed at 35 °C on potentiostat (Gamry Interface 1000, Gamry Instruments, Warminster, PA, USA). Ag/AgCl electrode (with saturated KCl electrolyte) was used as a reference, while graphite rod and cast iron were used as the counter and working electrodes, respectively. In three-electrode cell systems, we maintained enough (~1 mm) distance between the working and reference electrodes using a luggage probe to avoid a short circuit. Open circuit potential (OCP) was run for 30 min to stabilize the potential and maintain equilibrium between the working electrode and electrolyte. Electrochemical impedance was obtained at ±5 mV_rms_ AC potential amplitude vs. OCP in a frequency range of 100 kHz to 10 mHz. Tafel curves were obtained by polarizing the working electrode in the potential range of −1 to 1 V vs. OCP at a scan rate of 5 mVs^−1^.

Tribological studies were carried out by performing pin-on-disc tribo-tests of all the GCI samples with and without the addition of alloying elements by using Tribometer (MT/60/NI, Spain). Table 2 shows the conditions used to perform tribo-tests at humidity of 60% ± 5%.

The total load applied on the tungsten carbide ball during the tests was 20 N at a sliding distance of 30 m with a sliding time of 1800 s on the alloyed and un-alloyed GCI samples to analyze the wear properties. After measuring the cumulative wear volumes, specific wear rates were calculated by Equation (1):(1)$$K=\frac{V}{W.S}$$
where K is the specific wear rate (mm^3^/Nm); V is the cumulative wear volume (mm^3^); W is the normal load (N); and S is the sliding distance (m).

## 3. Results and Discussion

### 3.1. Microstructure

The analysis of the gray cast iron microstructure, including types, graphite content, and dimensions, as well as ferrite and pearlite area fraction determined using ImageJ software, is presented in Table 3. The addition of Ti and W notably altered the microstructure of pure gray cast iron, affecting the ferrite and pearlite ratios, grain size, and graphite characteristics. The reference Sample B-1 (pure cast iron) showed a type A microstructure with an average grain size of 378 ± 18 µm, consisting of 67% pearlite with 33% ferrite and 10.2 ± 2.3% graphite of an average graphite length of 31.9 ± 2.5 µm, as illustrated in Figure 2 Adding 0.612% Ti and 0.516% W to Sample B-1 (Sample B-2) demonstrated modification of the microstructure consisting of a small grain size, higher pearlite and graphite contents, along with lower maximum graphite length and ferrite fractions compared to reference Sample B-1. Furthermore, the tailored microstructure of the alloyed GCI Sample B-2 revealed the nucleation of E- and D-type graphite with A-type graphite flakes with a grain size of 363 ± 21 µm, pearlite volume fraction of 79%, ferrite volume fraction of 21%, uniformly distributed graphite content of 12.9 ± 2.1%, and average graphite length of 33.7 ± 2.4 µm (Figure 2).

Increasing the further concentration of Ti 0.762% and W 0.785% in Sample B-3 results in a similar trend in terms of the microstructural variation as of Sample B-2, but the graphite contents reduce to 12.2 ± 1.7%, as shown in Figure 3. Conversely, a higher concentration of the Ti 0.985% and W 1.351% contents in Sample B-4 demonstrated adverse effects compared to Sample B-3, but it has a similar microstructure to Sample B-1. The further increase in Ti and W did not significantly alter the microstructure of the base metal matrix (Sample B-1) and maintained the microstructure of dominant A-type graphite flakes with some B-type graphite flakes. The coarse microstructure of Sample B-4 shows a pearlite volume fraction of 74%, ferrite volume fraction of 26%, an average grain size of 358 ± 31 µm, graphite content of 13.3 ± 4.2%, and average graphite length of 36.3 ± 3.2 µm, as depicted in Figure 3. The microstructural investigation shows that Ti-W alloying has a strong influence on the type of and variation in the microstructure of GCI. Graphite formation occurs during the cooling of molten metal, which is then poured into a mold at room temperature [42]. The variation in the cooling temperature can cause the formation of various flaky graphites, such as B-, D-, and E-type (red encircled in Figure 2 and Figure 3) and microstructural structural morphology. The Ti alloy has a strong effect on the GCI solidification process, pearlite promotion, graphite formation, and refinement. During solidification, Ti quickly reacts with carbon, producing a plethora of tiny, evenly distributed titanium carbides. These carbides crucially influence the subsequent eutectic reaction stage, refine the grain size, increase the graphite contents, and promote its uniform distribution [43]. Therefore, during solidification in flaky GCI, Ti promotes the formation of different graphite flakes, pearlite nucleation, and a slower rate of ferrite formation due to its supercooling effects [44]. The other alloying element W refines the grain size and promotes pearlite in cast alloys [45,46].

The graphite flakes were classified as B-type, which are more rounded and refined, whereas D-type class graphite flakes develop under rapid cooling and tend to cluster in rosette-like patterns [47,48,49,50]. The E-type graphite flakes were very tiny and unevenly formed as a result of extremely rapid cooling rates combined with the significant carbide-forming activity of titanium and tungsten [50].

Based upon the microstructure analysis, it is concluded that the combination of Ti-W alloy has significantly modified the GCI microstructure.

The corresponding XRD spectra, EBSD images, and SEM morphology results are illustrated in Figure 4, Figure 5 and Figure 6, respectively. The crystalline phase of the base and alloyed GCI samples is illustrated in the XRD spectra of Figure 4, which validates the presence of Ti and W in the GCI Samples B-2, B-3, and B-4. The XRD spectrum of Sample B-1 showed one large and narrow intensity peak at 44.74°, presenting an austenite phase, and two more intensity peaks at 65.06° and 24.12°, presenting α-Fe (α-ferrite) and Fe_3_C (graphite) phases with the bod-centered cubic (BBC) in Sample B-1. Conversely, Samples B-2, B-3, and B-4 exhibited new peaks, suggesting the presence of austenite phase (γ-iron) and α-Fe (α-ferrite) phase, alongside some extra peaks that appeared at 25.98° 37.22°, and 75.6°. These extra peaks showed a gradual increase in the Ti and W contents in the produced alloyed cast irons (Samples B-2, B-3, and B-4), whereas the carbide in Sample B-1 was observed to be without Ti and W additives. It is observed that α-Fe (PDF card no. 06-0614), tungsten (PDF card no. 00-076-1877), and titanium carbide (PDF card no. 006.0614) were detected after adding alloying elements. These results are in agreement with the phases reported in the literature [51,52,53,54,55]. The formation mechanism of the alloying element in the metal matrix involved Ti-W reacting with carbon during solidification and formed the TiC and WC phase [56], which is counter-confirmed in the XRD crystallographic composition analysis. The concentration gradient of the Ti-W alloying elements is also observed at the periphery of the carbide phase within their respective 2θ ranges [57].

Further, to confirm and analyze the distribution of the alloying elements Ti and W throughout the gray cast iron, EDS maps were analyzed, as illustrated in Figure 5. A notable refinement in the pearlite structure was observed, marked by decreased pearlite clusters showing signs of refinement.

Additionally, the dissolution of W (tungsten) and Ti (titanium) into the pearlite and ferrite matrix contributed to solid solution strengthening (Figure 5). Ti and W were evenly distributed throughout the matrix though absent in graphitic flakes. The EDS maps revealed an increase in the Ti and W concentration in the matrix, correlating with their weight percentages increasing from 0.612 to 0.985 wt.% and 0.516 to 1.351 wt.% in gray cast iron, respectively.

### 3.2. Analysis of Alterations in Alloyed GCI Crystallographic Texture

The grain boundaries orientation map was measured by using the EBSD technique, as shown Figure 6. The multi-colored map shows the different grain directions as observed by distinct color boundaries that define the transition between the grains. The map indicates that alloying elements Ti and W in GCI affect the solidification, which results in the nucleation of different graphite flakes, grains, and sub-grains witnessed by the heterogeneous misorientation of the microstructure. Furthermore, Figure 6 also reveals the transition between high-angle grain boundaries (HAGBs) and low-angle grain boundaries (LAGBs), which confirms the refinement of the modified microstructure and grain size by the alloying of Ti-W. In the pristine Sample B-1 of GCI, 81.1% of LAGBs at 2–10° and 18.9% of HAGBs at >10° were identified by the misorientations, respectively. The alloyed Sample B-2 of GCI showed the evolution of the sub-grain and grain boundaries, with 62.9% of LAGBs at 2–5° and 37.1% of HAGBs at >5° [58]. The grain size and microstructure refinement started from LAGBs and was gradually merged into HAGBs, depicting the tailoring of the microstructure in alloyed GCI (Figure 6c,d). However, we did not observe the significant growth of HAGBs in Sample B-4, which is attributed to the retardation of grain growth [59]. The EBSD results proved that the alloying of Ti-W has tailored the microstructure of GCI, which caused the control of its mechanical, wear, and corrosion properties.

### 3.3. Mechanical Properties

Mechanical tests were performed to observe the effect of Ti and W additives on GCI. The results, including tensile strength, yield strength, and Brinell hardness, are presented in Table 4. The reference GCI sample (Sample B-1) without alloying exhibited the lowest mechanical properties, with a tensile strength of 346 ± 12 MPa, yield strength of 260 ± 3 MPa, and 213 ± 7 BHN Brinell hardness. This lower performance is attributed to its metal matrix composition due to the lowest pearlite volume (67%) and ferrite proportion (33%). In GCI, ferrite is known to be softer, whereas pearlite with layered cementite (Fe_3_C) is harder.

Upon introducing 0.612 wt.% Ti and 0.516 wt.% W to the matric of reference Sample B-1 and the alloyed sample, denoted as Sample B-2, there was an increase in pearlite to 79%, and ferrite was reduced to 21%. This resulted in an enhancement of the tensile strength by 4.62%, yield strength by 10%, and Brinell hardness by 13.14% compared to un-alloyed Sample B-1 [60]. The addition of Ti and W is reported to elevate the pearlite contents in the microstructure, enhancing the mechanical properties and reducing the graphite flake aspect ratio [61,62]. Further increasing the Ti and W content to 0.762 wt.% and 0.785 wt.%, respectively, Sample B-3 demonstrated an 8.38% increase in tensile strength, 19.23% increase in yield strength, and 27.69% increase in Brinell hardness compared to Sample B-1. This improvement aligns with the increase in pearlite to 87% and reduction in ferrite to 13% due to the higher Ti and W content [63,64]. Similarly, we have also observed the minimum values of grain size and maximum graphite length. All such microstructural refinement variations contributed towards the maximum mechanical properties of Sample B-3.

Sample B-4, showing even higher Ti (0.985 wt.%) and W (1.351 wt.%) content, showed a decline trend in terms of the mechanical property improvements, but this was better than Sample B-1. Sample B-4 demonstrated improvements in tensile strength of 1.4%, yield strength of 4.23%, and Brinell hardness of 14.08% compared to Sample B-1. This can be attributed to the 74% volume fraction of pearlite and 26% volume fraction of ferrite. Among all, Sample B-3 exhibited the highest tensile strength, yield strength, and Brinell hardness, establishing 0.762 wt.% Ti and 0.785 wt.% W as the optimal alloying element proportions. Exceeding these values may adversely affect the mechanical as well as microstructural properties. In conclusion, we can say that the improvement in the mechanical properties of alloyed GCI is mainly attributed to the modification of the microstructure through the increase in pearlite, refinement of grain size, and reduction in average graphite length.

The fracture characteristics during the tensile test were further examined using SEM (Figure 7). All samples, both base and alloyed, demonstrated brittle fracture behavior. De-cohesions observed along the ferrite/pearlite interface in Samples B-1, B-3, and B-4 indicate crack initiation sites. Moreover, cleavage facets indicative of low-energy brittle fracture were present across all the samples, with cleavage rupture being the dominant fracture mode under tensile stress [65,66].

### 3.4. Tribological Properties

The coefficient of friction (COF) for all the samples demonstrated (Figure 8a) that an initially upward increasing trend in the COF value was observed, and then afterward it gradually attained stability with time. The COF was eventually stabilized as a function of time, and the transition from instability to stability in the COF depends upon the nature of the frication pair contents. From the outset, the contents of the frication pair surface were characterized by the interlocking of ragged peaks, which elevates the COF significantly.

The wear rate and COF are significantly influenced by hard carbides, the pearlite ratio, and graphite flakes [67]. This correlation was evident from the COF and wear rates of both the base and alloyed GCI samples, which depend upon the percentage of pearlite volume and mechanical hardness, as shown in Figure 8a,b. The pearlite structure comprising cementite (Fe_3_C) constituents contributes to increasing the mechanical hardness and strength of GCI. Therefore, due to the lower pearlite contents and hardness, Sample B-1 showed the lowest COF and a higher wear rate (8.49 × 10^−3^ mm^3^/N.m), as demonstrated in Figure 8a,b. Conversely, the addition of alloying elements Ti-W improved the pearlite contents and hardness in Samples B-2 and B-3; consequently, improved wear resistance, a higher COF, and a lower wear rate, compared to Sample B-1, were observed. Furthermore, the increased Ti and W contents in Sample B-3 exhibited 87% of the pearlite volume percentage, and a value of 272 BHN of hardness was achieved. Therefore, the highest COF and lowest wear rate of 1.59 × 10^−3^ mm^3^/N.m were obtained among all the samples, which indicated the superior wear resistance of Sample B-3. Sample B-4 showed a declining trend in terms of COF and a slight rise in the wear rate of 3.21 × 10^−3^ mm^3^/N.m, However, it still outperformed Sample B-1.

Figure 9 illustrates the SEM images of the refined structure of alloyed GCI without defects or damages [68]. Fine graphite and hard carbides play a pivotal role in diminishing the wear rate. Despite these improvements, the effects of extensive ploughing and scratching are still visible on the substrate; especially, the wear track width reached up to 972.41 µm in Sample B-1, which indicates severe deformation, as shown in Figure 9. On the other hand, alloyed Sample B-2 showed a considerable decrease in wear width up to 471.07 µm, which significantly enhanced the wear resistance and substantial reduction in abrasion. Sample B-3 showed the enhanced impact of alloying elements to reduce wear track width from 972.41 µm to 438.08 µm, which is the maximum achievable reduction in the wear width. Therefore, it can be interpreted that the incorporation of W and Ti contributed towards optimal wear resistance owing to its high hardness value.

Additionally, Figure 10 revealed the surface roughness analysis performed by interferometer (VIS 633 nm), which gives an insight into the GCI wear properties through 3D profiling of the wear tracks. In Figure 8a,b, Sample B-1 demonstrated the highest surface roughness of 297.625 nm, in agreement with its COF and wear rate findings. The addition of Ti and W in Samples B-2 and B-3 led to an increase in the COF and decrease in the wear rate, accompanied by a gradual decrease in surface roughness. Sample B-3 showed the highest COF and lowest wear rate, wear width, and surface roughness, of 192.553 nm, among all the samples, as shown in Table 5. In conclusion, the decrease in the GCI wear rate increased its mechanical hardness. The higher mechanical hardness caused an improvement in the wear resistance due to anti-surface deformation. Therefore, the excellent wear resistance of alloyed GCI was due to its surface hardness caused by the refined grain size and higher pearlite volume percentage. The surface morphology of alloyed GCI also validated this phenomenon.

### 3.5. Corrosion Analysis

#### 3.5.1. Potentiodynamic Polarization Analysis

The potentiodynamic polarization (PDP) of alloyed and un-alloyed GCI is analyzed in seawater conditions to investigate the improvement in the electrochemical properties. The corrosion potential (V), current densities (A/cm^2^), and corrosion rates (mpy) were derived from PDP experimental curves (Figure 11). Detailed data of the abovementioned experimental curves are presented in Table 6. In GCI, the corrosion potential is determined through electrochemical analysis, indicating interplay between anodic and cathodic reactions; the anodic reaction involves the dissolution of iron, and the cathodic reaction typically entails hydrogen evolution.

A comparative analysis of the corrosion rates between pure and alloyed GCI reveals the impact of varying alloy concentrations on the corrosion resistance, which provides insights into determining the optimal concentration of alloying elements for the enhancement of corrosion resistance. Titanium is the main constituent which presents as an alloying element and forms titanium chloride (TiCl_3_), which hinders the cathodic reaction. This shows that the corrosion reaction is inhibited by the formation of TiCl_3_. By extrapolating the linear segments of the cathodic and anodic branches in the polarization curves to their intersection, we obtained the corrosion potential (E_-_corr) and corrosion current density (I_-_corr) [69,70,71].

The samples exhibited varied corrosion potentials and current densities. Notably, Sample B-3 demonstrated the highest corrosion potential, moving in a positive or noble direction, indicative of superior corrosion resistance. The reference GCI sample with high carbon contents and an absence of alloying elements showed lower corrosion resistance, primarily due to the graphitization of the metal matrix, which undermines its protective properties. The enhanced corrosion resistance observed in the alloyed samples highlights the significance of alloying to improve these properties [72]. Specifically, Sample B-3 exhibited the maximum corrosion potential (E_-_corr) −0.4813 V and a corrosion current density (I_-_corr) of 2290 A/cm^2^ [73]. The negative values of the current densities are also reported in a previous study [74].

#### 3.5.2. Electrochemical Impedance Spectroscopy

To assess the electrochemical properties of GCI alloyed with a different weight% of Ti-W elements, electrochemical impedance spectroscopy (EIS) analysis was conducted. Figure 12 presents the Nyquist plots of Samples B-1, B-2, B-3, and B-4, with varying concentrations of alloying elements. These plots feature the capacitive loops at high-frequency ranges, indicative of charge transfer resistance in the analyzed samples. Notably, the capacitive loop of Sample B-3, with 0.762% Ti and 0.785% W, is more pronounced compared to other samples, demonstrating the higher charge transfer resistance of Sample B-3. Higher charge transfer resistance is typically associated with enhanced corrosion resistance [75,76,77,78].

Bode impedance plots, illustrating the behavior of Samples B-1, B-2, B-3, and B-4, corroborate these findings. As shown in Figure 13, these plots reveal resistive behavior at high frequencies and capacitive behavior at medium-to-low frequencies. The impedance curves are similar across all samples, differing primarily in the magnitude of the bode impedance. At a high-frequency range (10^3^ to 10^5^ Hz), the curves exhibit resistive behavior indicated by their parallel alignment with the frequency axis. Conversely, at medium frequencies usually below 10^3^, a decrease in frequency correlates with increased capacitive reactance, signifying more pronounced capacitive behavior at lower frequencies. Consequently, Sample B-3 demonstrates the highest bode impedance of ~2290 Ohm.cm^2^, further confirming its superior corrosion resistance, which can be attributed to alloying elements [79,80]. Both PDP and EIS analysis confirm that Sample B-3 containing 0.762% Ti and 0.785% W by weight% used in GCI exhibits significantly improved corrosion resistance in the seawater concentration. This improvement is a direct consequence of the specific concentrations of Ti and W used in GCI.

## 4. Conclusions

The alloying effect of Ti and W on the microstructural, mechanical, wear, and corrosion properties of GCI has been thoroughly investigated in this work. The main results are as follows:(1)The addition of Ti and W has transformed the A-type graphite flaky microstructure into B + E + D-type, which resulted in an improvement in the pearlite volume percentage and the refinement of grain size. The improvement in the mechanical properties can be attributed to the enhancement of the volume fractions of pearlite from 67% to 87%, reducing the average grain size from 378 µm to 349 µm and maximum graphite length from 356 µm to 297 µm.(2)The hardness of the alloyed GCI sample was improved from 213 BHN to 272 BHN, which demonstrated a reduced wear rate from 8.49 × 10^−3^ mm^3^/N·m to 1.59 × 10^−3^ mm^3^/N·m and surface roughness from 297.625 nm to 192.553 nm, which indicated a significant improvement in the wear resistance of alloyed GCI.(3)The lower corrosion resistance, ~1545 Ohm.cm^2^, of the pristine GCI sample with a coarse grain size and high ferrite volume fraction enhanced the formation of graphitization in the metal matrix. The alloying of tungsten and titanium enhanced the corrosion resistance to ~2290 Ohm·cm^2^, owning to the refinement of the microstructure, improvement in the pearlites volume fraction, and scarcity of ferrites in GCI.(4)The crystallographic study showed the misoriented microstructure with grain and sub-grain boundaries distributed at high and low angles. A successfully tailored microstructure of GCI by alloying Ti-W was observed due to the nucleation of grain and sub-grain boundaries, which improved the mechanical, wear, and corrosion properties.

## Figures and Tables

**Figure 1 materials-17-02468-f001:**
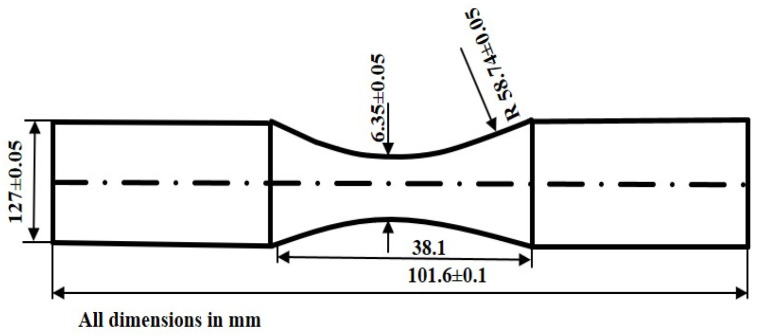
Schematic diagram of tensile test sample.

**Figure 2 materials-17-02468-f002:**
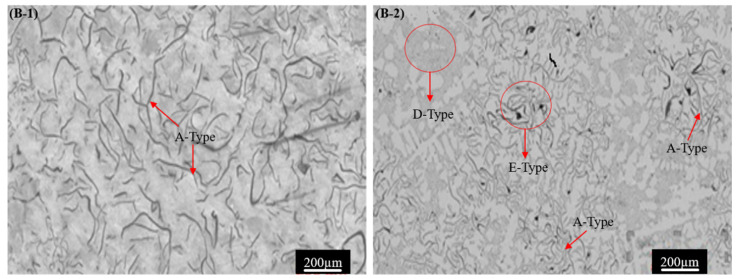
Microstructure of GCI Samples B-1 and B-2.

**Figure 3 materials-17-02468-f003:**
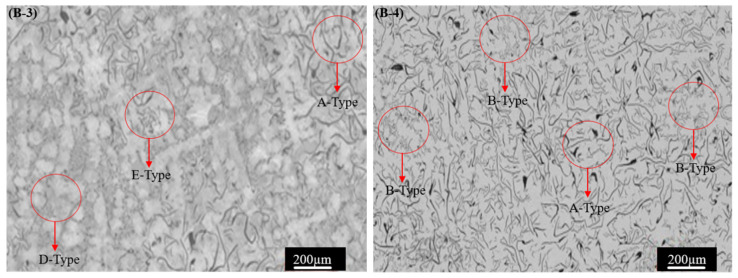
Microstructure of GCI Samples B-3 and B-4.

**Figure 4 materials-17-02468-f004:**
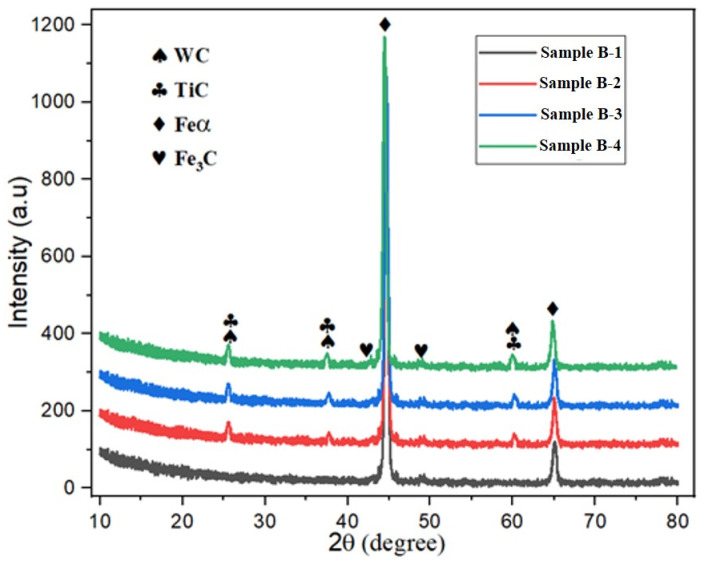
XRD phase analysis of Samples A, B, C, and D.

**Figure 5 materials-17-02468-f005:**
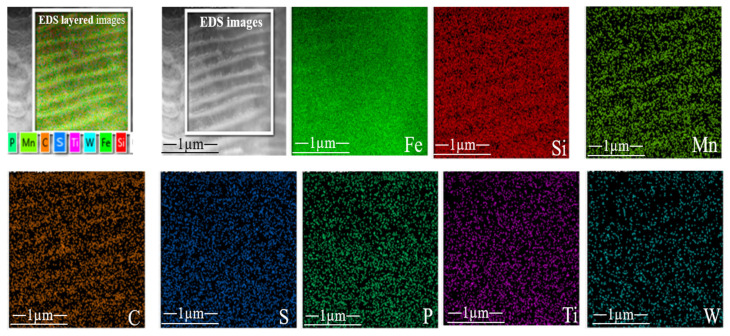
STEM micrograph of pearlite and ferrite in Sample B-3 and corresponding EDS element chemical distribution diagram and energy spectrum analysis.

**Figure 6 materials-17-02468-f006:**
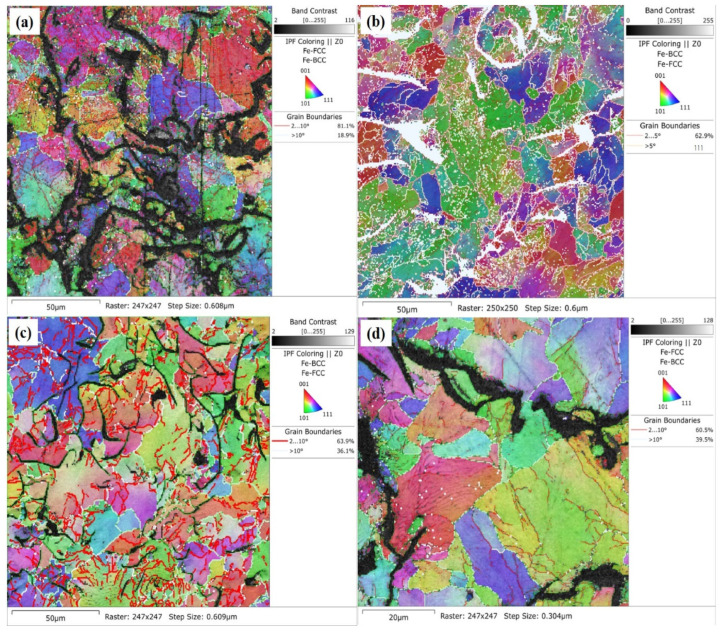
EBSD orientation map of Sample B-1 (**a**), B-2 (**b**), B-3 (**c**), and B-4 (**d**) grain boundaries.

**Figure 7 materials-17-02468-f007:**
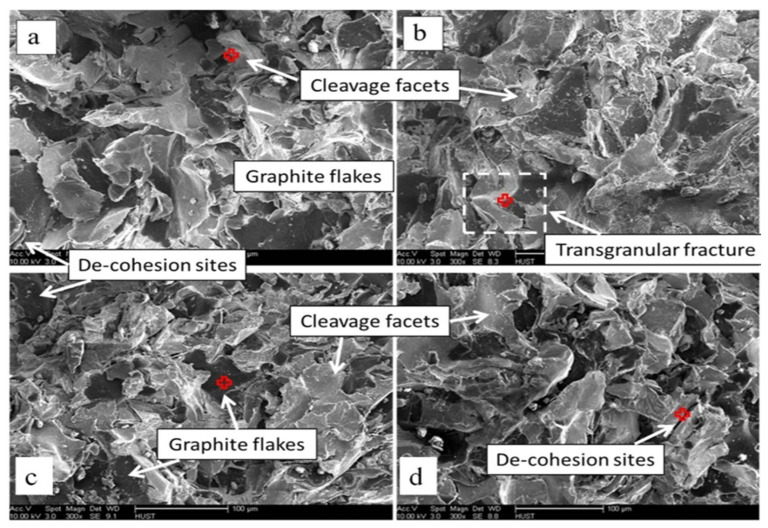
Fractography of (**a**) Sample B-1, (**b**) Sample B-2, (**c**) Sample B-3, and (**d**) Sample B-4.

**Figure 8 materials-17-02468-f008:**
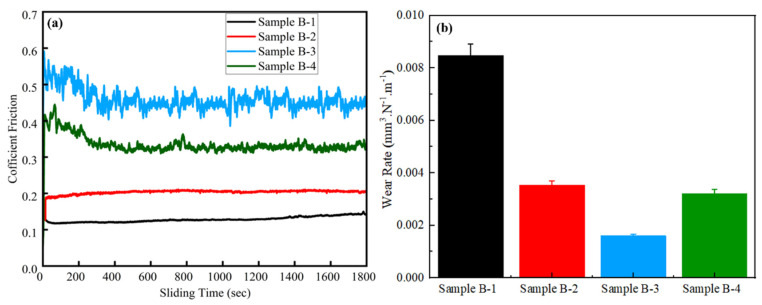
Samples B-1, B-2, B-3, and B-4; (**a**) coefficient of friction and (**b**) wear rate.

**Figure 9 materials-17-02468-f009:**
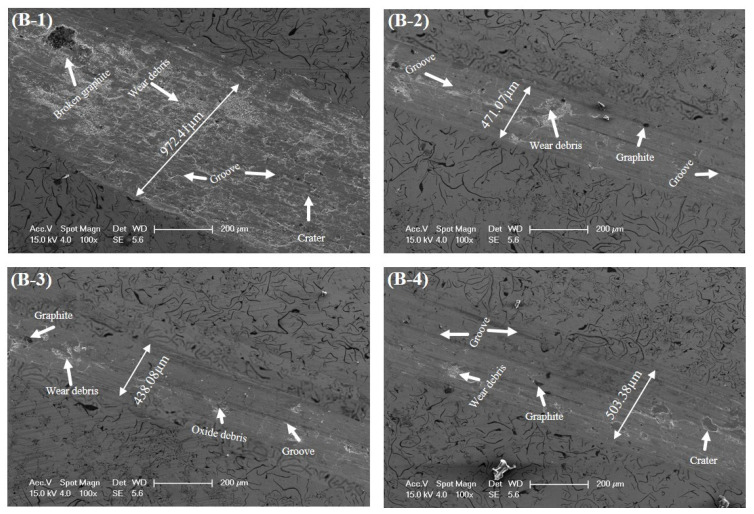
SEM micrographs of worn surface of Samples B-1, B-2, B-3, and B-4.

**Figure 10 materials-17-02468-f010:**
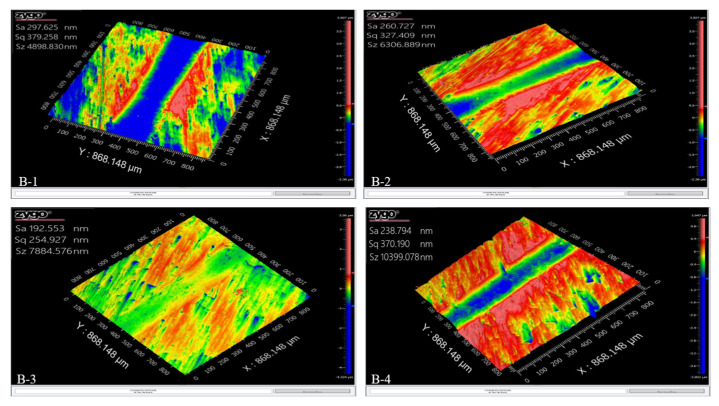
Three-dimensional profiles of wear tracks of Samples B-1, B-2, B-3, and B-4.

**Figure 11 materials-17-02468-f011:**
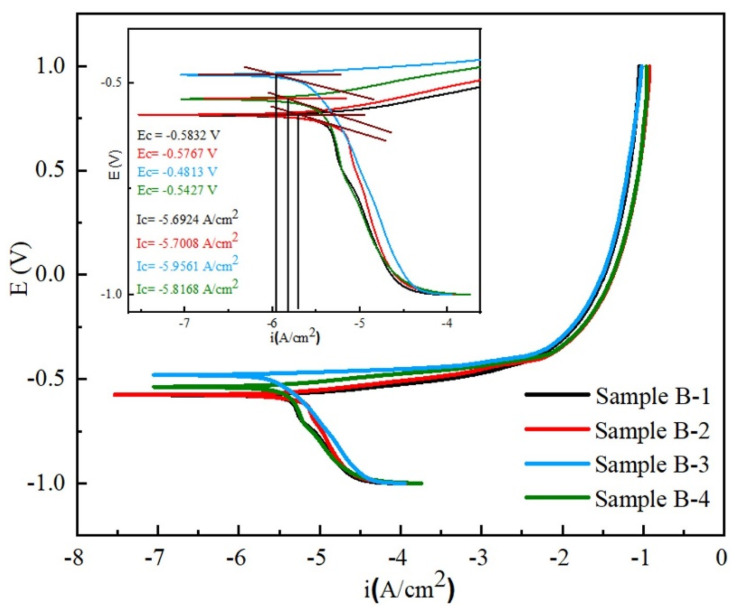
Potentiodynamic polarization curves of Samples B-1, B-2, B-3, and B-4.

**Figure 12 materials-17-02468-f012:**
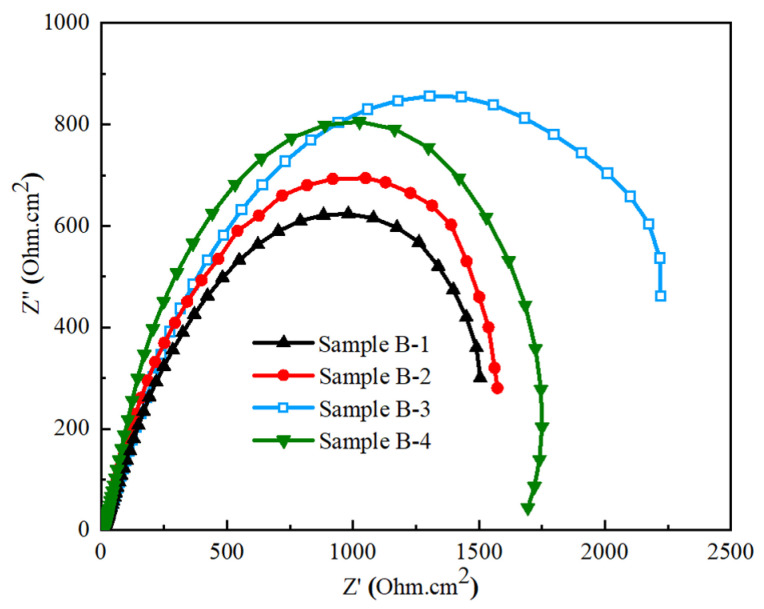
Nyquist plots of Samples B-1, B-2, B-3, and B-4.

**Figure 13 materials-17-02468-f013:**
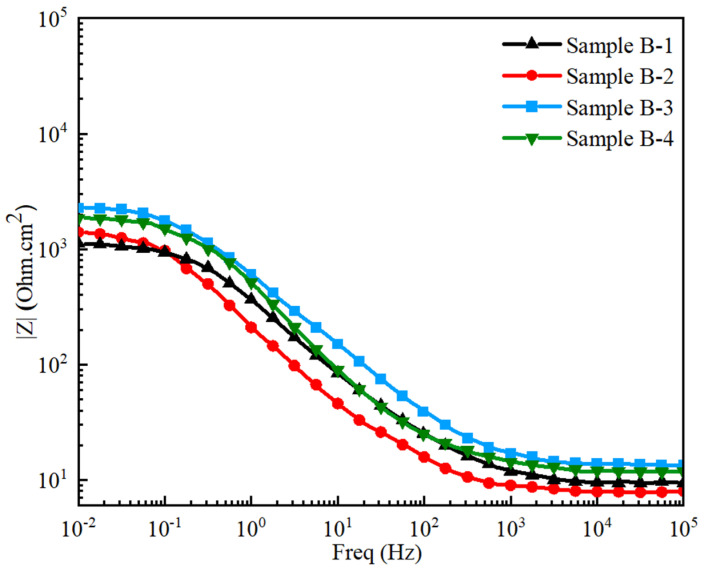
Bode impedance plots of GCI Samples B-1, B-2, B-3, and B-4.

**Table 1 materials-17-02468-t001:** Base metal composition with and without alloying elements Ti and W (wt.%).

Sample	C	Si	Mn	P	S	Ti	W
**B-1**	3.321	2.206	0.567	0.025	0.081	0	0
**B-2**	3.435	2.109	0.542	0.031	0.082	0.612	0.516
**B-3**	3.395	2.157	0.576	0.029	0.092	0.762	0.785
**B-4**	3.294	2.125	0.593	0.024	0.089	0.985	1.351

**Table 2 materials-17-02468-t002:** Tribological test parameters of GCI.

Parameter	Value	Parameter	Value
Applied load (N)	20	Revolution (rpm)	50
Sliding time (s)	1800	Wear track radius (mm)	3
Atmosphere	Dry	Temperature °C	30 °C ± 5 °C

**Table 3 materials-17-02468-t003:** Microstructure analysis of Samples B-1, B-2, B-3, and D-4.

Sample #	Structure Type	Graphite Content, %	Average Graphite Length, µm	Max Graphite Length, µm	Average Grain Size, µm	Pearlite and Ferrite, %
**B-1**	Type A	10.2 ± 2.3	31.9 ± 2.5	356 ± 31	378 ± 18	67/33
**B-2**	Type A + D + E	12.9 ± 2.1	33.7 ± 2.4	319 ± 19	363 ± 21	79/21
**B-3**	Type A + D + E	12.2 ± 1.7	35.2 ± 3.7	297 ± 16	349 ± 19	87/13
**B-4**	Type A + B	13.3 ± 4.2	36.3 ± 3.2	324 ± 09	358 ± 31	74/26

**Table 4 materials-17-02468-t004:** Mechanical properties of alloyed and un-alloyed GCI samples.

Sample #	Tensile Strength (MPa)	Yield Strength (MPa)	Brinell Hardness (BHN)
**B-1**	346 ± 12	260 ± 3	213 ± 7
**B-2**	362 ± 9	286± 11	241 ± 4
**B-3**	375 ± 7	310± 2	272 ± 8
**B-4**	351 ± 10	271± 6	243 ± 3

**Table 5 materials-17-02468-t005:** The compendium of tribological experimental results.

Sample	COF	Wear Rate (mm^3^/N·m)	Wear Width (µm)	Surface Roughness (nm)
**B-1**	0.12	8.49 × 10^−3^	972.41	297.625
**B-2**	0.19	5.52 × 10^−3^	471.07	260.727
**B-3**	0.51	1.59 × 10^−3^	438.08	192.55
**B-4**	0.40	3.21 × 10^−3^	503.38	238.794

**Table 6 materials-17-02468-t006:** Kinetic parameters of Samples B-1, B-2, B-3, and B-4.

Sample ID	i (A/cm^2^)	E_corr_ (V)	R_P_ (Ohm·cm^2^)
**B-1**	−5.6924	−0.5832	1545
**B-2**	−5.7008	−0.5767	1654
**B-3**	−5.9561	−0.4813	2290
**B-4**	−5.8168	−0.5427	1856

## Data Availability

Data are contained within the article.

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
