# Peer review of "Improvements in Wear and Corrosion Resistance of Ti-W-Alloyed Gray Cast Iron by Tailoring Its Microstructural Properties"

_materials, 2024, doi:10.3390/ma17102468_

Round 1
Reviewer 1 Report
Comments and Suggestions for Authors
“Improvements in Wear-and-Corrosion Resistance of Ti-W Alloyed Gray Cast Iron by Tailoring its Microstructural Properties”
Authors:
Abdul Razaq, Peng Yu, Adnan Raza Khan, Xiao-yuan Ji, Ya-jun Yin, Jian-xin Zhou, Taher A. and ehabeldeen
In the paper, authors wanted to improve wear and corrosion resistance of gray cast iron by addition of titanium and tungsten to it. It was expected that it caused changes in the microstructure of gray cast iron as well as its wear and corrosion resistance would be higher.
The paper is interesting, but there is no information possible use of this cast iron.
Where and under what conditions could it be tested?
Author Response
- Additional explanations and regulations should be added to help readers understand the criteria for classifying types A to E of GCI.
Response: Thank you for this recommendation, the query was addressed at appropriate places in section 3.1 para#2 &3 and section 3.2, para#1.
- Labeling samples as A to D may cause confusion with GCI types. Sample names should be represented differently.
Response: Thank you for your valuable suggestions. We have updated the Sample labels in the manuscript. The updated sample names are from B-1 to B-4.
- It should be confirmed how the structure type classifications in Table 3 are represented in Figures 2 and 3.
Response: The graphite flakes structure type classifications (A, B, D and E) can be identified as red circles in Figures 2 and 3 of the manuscript, which were mentioned in Table 3.
- The colors used to differentiate samples in each graph from Figure 4 to Figure 13 should be standardized.
Response: As suggested by the reviewer, the colors has been standardized as desired and uniform standard has been implemented from figure 4 to 13 in the manuscript.
- Discrepancies are observed in the mechanical properties results based on Sample C (B-3) in comparison to the microstructural analysis outcomes presented in Table 3. A detailed explanation should be provided regarding the specific reasons for the increases and decreases observed due to the addition of Ti and W.
Response: The existence of co-relation between microstructural and mechanical properties of Sample C (B-3) has established and elaborated with necessary details in the manuscript Section 3.3. Hopefully, it will remove the ambiguity in correlation between sample C mechanical properties and its microstructure analysis listed Table 3.
- The corrosion potential values in Table 5 (Table 6), representing the electrochemical test results, do not align with the graph in Fig. 11. Data verification and figure adjustment are necessary.
Response: Authors are thankful for your valuable comment and suggestion regarding improving the quality of research article. We have revised the figure 11 and the manuscript has been updated with new values of corrosion potentials and corrosion current densities as well as incorporated in the figure 11 as desired.
- An explanation regarding which alloy constituents influence the change in corrosion characteristics should be provided.
Response: Titanium is the main constituent which present as an alloying element and forms the titanium chloride (TiCl3) which hinders the cathodic reaction. This shows that corrosion reaction is inhibited by formation of TiCl3 [70]. ( An approach for high stability TiO2 in strong acidic environment) by Badar Minhas).
- Unlike Sample C (B-3), the current density of B-2 and B-4 was evaluated to increase compared to B-1. What are the rea
Response: Kindly refer to figure 11, where values of current densities for all samples has been revised. It is evident from the keen observation of figure 11 that sample B-3 shows the highest current density (−0.4813) as compared to samples B-1, B-2 and B-4.
- The EIS data in Table 6 does not correspond with the graph in Fig. 12. Data verification and figure adjustment are required.
Response: EIS spectra of all the samples has been re-evaluated, shown in the figure 12. The investigation results showed that corrosion resistance values are in good agreement with Table 6.

Reviewer 2 Report
Comments and Suggestions for Authors
There are some correction to be made.
1. Main question: A new composition of gray cast iron with better properties. 2. In this domain, new ideas and research are rare. About 30 years ago there were more studies, with classical solutions. 3. The use of Titanium and Tungsten seems to have an improvement in gray cast iron properties.
4. The research methodology is normal and investigations were made with modern technology.
5. Try to use the correct symbol for multiplication, even for measurement units.
Comments on the Quality of English Language
Some errors (missing letters) should be corrected.
Author Response
1. Try to use the correct symbol for multiplication, even for measurement units. Response: As per suggestion of reviewer, we have revised and updated the manuscript at relevant places.

Reviewer 3 Report
Comments and Suggestions for Authors
Review
1. Additional explanations and regulations should be added to help readers understand the criteria for classifying types A to E of GCI.
2. Labeling samples as A to D may cause confusion with GCI types. Sample names should be represented differently.
3. It should be confirmed how the structure type classifications in Table 3 are represented in Figures 2 and 3.
4. The colors used to differentiate samples in each graph from Figure 4 to Figure 13 should be standardized.
5. Discrepancies are observed in the mechanical properties results based on Sample C in comparison to the microstructural analysis outcomes presented in Table 3. A detailed explanation should be provided regarding the specific reasons for the increases and decreases observed due to the addition of Ti and W.
6. The corrosion potential values in Table 5, representing the electrochemical test results, do not align with the graph in Fig. 11. Data verification and figure adjustment are necessary.
7. An explanation regarding which alloy constituents influence the change in corrosion characteristics should be provided.
8. Unlike Sample C, the current density of B and D was evaluated to increase compared to A. What are the reasons for this?
9. The EIS data in Table 5 does not correspond with the graph in Fig. 12. Data verification and figure adjustment are required.
Author Response
- In the paper, the authors wanted to improve wear and corrosion resistance of gray cast iron by addition of titanium and tungsten to it. It was expected that it caused changes in the microstructure of gray cast iron as well as its wear and corrosion resistance would be higher. The paper is interesting, but there is no information possible use of this cast iron. Where and under what conditions could it be tested?
Response: Introduction section was updated by incorporating the applications and benefits of alloyed GCI to be used under harsh environmental conditions and elevated mechanical loads. Our research results are based upon physical and experimental testing of GCI samples conducted at available lab facility to mimic the extreme service conditions.

Round 2
Reviewer 3 Report
Comments and Suggestions for Authors
Accept in present form